# Post-Harvest Contamination with Mycotoxins in the Context of the Geographic and Agroclimatic Conditions in Romania

**DOI:** 10.3390/toxins10120533

**Published:** 2018-12-13

**Authors:** Valeria Gagiu, Elena Mateescu, Ileana Armeanu, Alina Alexandra Dobre, Irina Smeu, Mirela Elena Cucu, Oana Alexandra Oprea, Enuta Iorga, Nastasia Belc

**Affiliations:** 1National Research & Development Institute for Food Bioresources—IBA Bucharest, 5 Baneasa Ancuta Street, 2nd district, Bucharest 020323, Romania; armeanuileana@yahoo.com (I.A.); alina.dobre@bioresurse.ro (A.A.D.); irina.smeu@bioresurse.ro (I.S.); mirela.cucu@bioresurse.ro (M.E.C.); enutai@yahoo.com (E.I.); nastasia.belc@bioresurse.ro (N.B.); 2National Meteorological Administration (METEO—Romania), 97 Bucuresti‒Ploiesti Street, 1st district, Bucharest 013686, Romania; elena.mateescu@meteoromania.ro (E.M.); oprea@meteoromania.ro (O.A.O.)

**Keywords:** Mycotoxin, post-harvest, geographic position, agroclimatic conditions, climate change, Romania, southeast Europe

## Abstract

This study aimed to assess post-harvest contamination with mycotoxins in the context of the geographic and agroclimatic conditions in Romania in 2012–2015, a period that was characterized by extreme meteorological events and the effects of climate change. The samples were randomly sampled from five agricultural regions of Romania and analyzed for mycotoxins by enzyme-linked immunosorbent assay. An SPSS analysis was done to explore correlations between mycotoxins (deoxynivalenol—DON, aflatoxins—AF, ochratoxin A—OTA, zearalenone—ZEA), product types (raw cereal, processed cereal, cereal-based food), geographic coordinates (latitude, longitude, agricultural region), and agroclimatic factors (air temperature, precipitation, soil moisture reserve, aridity index, soil type). In the southeast part of the Southern Plain and Dobrogea (Baragan Plain, located at 44–45° N, 26–27° E), contamination with AF and OTA was higher in raw and processed cereals (maize in silo, silo corn germs) in the dry years (2012 and 2013), and contamination with DON was high in processed cereals (wheat flour type 450) in the rainy year (2014). DON and OTA contamination were significantly correlated with cumulative precipitation in all years, while AF and ZEA contamination were non-significantly correlated with climatic factors and aridity indices. The distribution of mycotoxins by product type and the non-robust correlations between post-harvest mycotoxins and agrometeorological factors could be explained by the use of quality management systems that control cereal at warehouse receptions, performant processing technologies, and the quality of storage spaces of agri-food companies. The Baragan Plain is Romania’s most sensitive area to the predicted climate change in southeast Europe, which may be associated with its increased cereal contamination with AF and OTA.

## 1. Introduction

Cereal contamination occurs in the field under the influence of agroclimatic conditions that are favorable to fungal attack and mycotoxin production, and may continue post-harvest in the storage–processing–commercialization chain, thus resulting in financial losses and food safety and security issues [1,2,3,4]. Awareness of the effects of current and future agroclimatic conditions as well as the implementation of integrated mycotoxin management systems are issues of significant concern at both the national and global levels as a result of grain exports by the big traders.

Romania is located in southeast Europe between latitudes 43° N and 48° N and longitudes 20° E and 29° E, and has a humid temperate continental climate with hot summers in parts of the Southern country [5,6]. Due to its agroclimatic conditions, soil types, and agricultural technologies, Romania is a significant cereal producer at the global level. In the period of 1961–2016, it belonged to the first group for maize (8,800,672.82 t) and wheat (5,724,718.45 t), the second group for soybeans (180,864.95 t), the third group for triticale (104,945 t), and the fourth group for sorghum (15,539.54 t) [7]. From a capability point of view, arable soils in Romania are classified as very fertile (chernozem), fertile (phaeozem), or moderately fertile (luvisol) [8]. Since chernozem and phaeozem have high agroclimatic aridity indices [6,9], these soils require irrigation—especially in dry years—to sustain the fertility, productivity, and quality of crops. The agricultural area of Romania is divided into regions with different agroclimatic conditions due to relief forms and significant differences in the quantity and quality of cereal crops: Zone 1—Southern Plain, Oltenia Plain, and Western Plain (moderate temperate continental climate, chernozem); zone 2—Moldavian Plain and Dobrogea Plain (arid temperate continental climate, chernozem); and zone 3—Transylvania and the Southern Hilly Area (wet temperate continental climate, luvisol).

Failure to respect agricultural technologies and the existence of extreme meteorological events (high temperature, drought, heavy rain) adversely affects crop quantity and quality, favoring fungal attack and mycotoxin production. Until 2012, most Romanian studies reported the incidence and range of mycotoxins in cereals, food, and feed in separate regions, without an assessment of these factors in the geographic and agroclimatic contexts [10,11,12,13,14,15,16]. In-field contamination with deoxynivalenol was evaluated in wheat, rye, and triticale crops from 2012 to 2016, and in the wheat chain in 2015, taking into account the geographic positions and pedoclimatic conditions of the agricultural regions [17,18].

In Romania, the climate scenario is forecasted to involve a rise in temperature by 3 °C–5 °C in the north and 4 °C–5 °C in the south by 2100, with rainfall reduction and drought especially in the south, followed by a decline in crop yields of up to 20% [19,20,21,22]. Considering the current and future climate conditions, the socio-economic factors, and the political context as a European Union member country, it was predicted that Romania will present a medium climate risk index and an average vulnerability to climate change among the countries of southeast Europe [23,24]. 

The aim of this article was to assess post-harvest contamination with mycotoxins in the context of the geographic and agroclimatic conditions of Romania and climate change in southeast Europe. The research objectives were (a) to quantify mycotoxins in post-harvest samples in the period of 2012–2015, taking into account extreme meteorological events; (b) to statistically analyze post-harvest contamination with mycotoxins by year, product type, geographic coordinates, and agroclimatic data; and (c) to determine possible trends in mycotoxin contamination in Romania and southeast Europe in the context of climate change. This represents the first time that post-harvest contamination with mycotoxins has been investigated in Romania and southeast Europe in this context, and the approach used in this study can be applied in any country or even at the continental level along the agri-food chain. This approach was chosen because it correlates mycotoxin contamination with local agroclimatic factors, and because it can be used by risk managers in agri-food companies and integrated into the control systems of competent authorities. 

## 2. Results

### 2.1. Agrometeorological Factors

In Romania, the growth of cereal crops depends on the value of agrometeorological factors during periods of maximum water consumption—May–June for winter wheat and June–August for maize. During the agricultural period of September 2011 to August 2015, the air temperature, precipitation, and soil moisture reserve varied between agricultural regions characterized by different relief forms. Appendix A provide the results of the ANOVA and Kruskal–Wallis nonparametric test of the physical context (agroclimatic factors: Air temperature, precipitation, water reserve in the soil) in which mycotoxin contamination occurred, also indicating grouping by years and locations.

#### 2.1.1. Air Temperature

The average annual air temperature showed very significant differences *(p* < 0.001) by region and year (Appendix A). The highest air temperatures were recorded in the warm and dry Southern regions (Oltenia Plain, Southern Plain and Dobrogea), followed by the humid and temperate regions (Southern Hilly Area, Western Plain) and colder regions (Moldavia, Transylvania) (Appendix A). The year 2012 was the warmest and was significantly different from the other years. The next warmest year was 2015, while the differences in average air temperature between 2013 and 2014 were non-significant (Appendix A).

#### 2.1.2. Precipitation

The average cumulative precipitation in March–August showed non-significant differences between regions, but very significant differences between years (*p* < 0.001) (Appendix A). The highest precipitation was recorded in Southern Romania (Southern Hilly Area, Southern Plain, and Dobrogea) in 2014, and the lowest precipitation was also recorded in Southern Romania (Southern Plain and Dobrogea) in 2012 and 2015. The year, 2014, was the rainiest year, followed by 2013, and finally 2012 and 2015 between which the differences were non-significant (Appendix A).

#### 2.1.3. Soil Moisture Reserve

The average soil moisture reserve was not significantly difference between regions or years according to the ANOVA test.

However, after reanalyzing the agrometeorological factors with the Kruskal–Wallis non-parametric test for independent samples, significant differences were found for the distribution of temperature and cumulative precipitation between regions and the distribution of the soil moisture reserve between regions and years (Figure 1, Appendix A).

The years, 2012 and 2015, were part of a series of extreme dry years that occurred in the 21st century in Romania (2002, 2003, 2007, 2012, and 2015), in which there were more than 60 days with maximum temperatures ≥32 °C, insufficient rainfall (≤200 mm), and a soil water deficit, especially in the southeast part of the country (Baragan Plain) during the period of June to August [25].

### 2.2. Post-Harvest Contamination with Mycotoxins

#### 2.2.1. Deoxynivalenol (DON)

Post-harvest contamination with DON had an incidence rate of 42.9% (39/91) and varied between <18.5 and 1269.94 µg/kg (mean 82.39 ± 158.42 µg/kg, median <18.5 µg/kg); the incidence of samples above maximum level was 1.1% (1/91). The highest DON level was detected in a sample of wheat flour type 450 (processed cereals, in the year 2015) obtained from in-field contaminated wheat in the rainy year of 2014 after 11 months of storage (Figure 2A and Figure 3A). The incidence of DON-positive samples was high in the cold and wet areas of Transylvania (36.4%) and in the Southern Hilly Area (51.4%), as well as in the warm and arid Southern Plain and Dobrogea due to the abundant rainfall in 2014 (43.3%) (Figure 3A). The highest DON incidence was recorded in the year, 2014, which was very rainy in May–July (55.6%), followed by 2013 and 2012, which were rainy in May–June and very dry in July–August (41.2%, 38.5%) (Figure 3A). The processed cereal (wheat flour type 450) with DON above maximum level was sampled in May 2015 from the Southern Plain and Dobrogea (Chirnogi, Calarasi county) (Figure 4) and was possibly obtained from wheat contaminated with 1025 µg/kg DON in the rainy year of 2014 [26]. The contamination was caused by the abundant rainfall from May to July 2014 in the south; however, the region is not usually favorable to *Fusarium* attack and DON production due to its agroclimatic conditions (arid temperate continental climate, chernozem soil with high aridity indices). As such, only in-field cereals subjected to extreme weather conditions were affected [15,17,19,26].

#### 2.2.2. Total aflatoxins (AF)

Post-harvest contamination with AF had an incidence rate of 45.4% (44/97), which varied between <1.75 and 82.94 µg/kg (mean 3.85 ± 14.80 µg/kg, median <1.75 µg/kg); the incidence of samples above the maximum level was 4.1% (4/97). The highest AF levels were detected in raw cereal (2012: Maize grains in silos—max. 82.94 µg/kg; 2013: Maize grains—max. 79.99 µg/kg) and processed cereals (2012: Maize extract—max. 4.53 µg/kg; 2013: Wheat flour type 1350—max. 8.34 µg/kg) (Figure 2B and Figure 3B). The detection of AF in maize grains in silos from October 2012 and August 2013 showed that contamination had occurred in-field in the dry years of 2012 and 2013. The incidence of AF was higher in the dry areas of Moldavia (12.5% positive samples; 12.5% samples above maximum level; max. 4.53 µg/kg) and in the Southern Plain and Dobrogea (8.4%; max. 82.94 µg/kg), while its incidence was sporadic in the cold and humid areas of Transylvania and in the Southern Hilly Area due to the local conditions in each year (Figure 2A,B). The highest incidence of AF above the maximum level was recorded in 2012 (7.7%; max. 82.94 µg/kg) and 2013 (7.3%; max. 79.99 µg/kg), which had low rainfall and high temperatures in the critical period of June–August. There was no sample above maximum level in 2014 and 2015 (Figure 2B and Figure 3B). The raw and processed cereals (maize grain in silos, silo-corn germs, maize extract) with AF above maximum levels were sampled from the Southern Plain and Dobrogea (Ianca, Braila county; Tandarei, Ialomita county) and Moldavia (Iasi, Iasi county) (Figure 4), where the agroclimatic conditions are favorable to *Aspergillus* sp. and AF production (arid temperate continental climate, chernozem soil with high aridity indices).

#### 2.2.3. Ochratoxin A (OTA)

Post-harvest contamination with OTA had an incidence rate of 6.8% (4/59), which varied between <2.50 and 6.72 µg/kg (mean 2.70 ± 0.43 µg/kg, median <2.50 µg/kg); the incidence of samples above the maximum level was 6.8% (4/59). The highest OTA levels were detected in processed cereals (2012: Maize grain in silos—max. 3.39 µg/kg; 2014: Three samples of silo-corn germs—max. 5.64 µg/kg, 6.24 µg/kg, and 6.72 µg/kg) (Figure 3C). OTA only exceeded the maximum level in samples from the warm and arid Southern Plain and Dobrogea (10.0% positive samples; 10% samples above maximum level; max. 6.72 µg/kg) (Figure 2C). The highest incidence of OTA above maximum level was recorded in the dry year of 2012 (8.3%; max. 3.39 µg/kg) and the rainy year of 2014 (18.8% positive samples; 18.8% samples above maximum level; max. 6.72 µg/kg) (Figure 2C). However, the silo-corn germ samples from 2014 were obtained from corn contaminated in-field in the dry year of 2013 and stored for four months. The processed cereals (maize grains in silos; silo-corn germs) with OTA above the maximum level were sampled from the Southern Plain and Dobrogea (Faurei, Braila county; Tandarei, Ialomita county) (Figure 4) where, as previously mentioned, the agroclimatic conditions are favorable to *Aspergillus* sp. and OTA production.

#### 2.2.4. Zearalenone (ZEA)

Post-harvest contamination with ZEA had an incidence rate of 7.1% (6/84), which varied between <1.75 and 7.05 µg/kg (mean 1.92 ± 0.22 µg/kg, median NA); the incidence of samples above the maximum level was 0% (0/84). The highest levels were detected in processed cereals in the year, 2014 (silo-corn germs—max. 4.65 µg/kg and 7.05 µg/kg), with both samples being from maize 2013 (Figure 2D). ZEA contamination was low in the Southern Hilly Area (5.6%; max. 2.94 µg/kg), which has a humid temperate continental climate, and the Southern Plain and Dobrogea (13.8%; max. 7.05 µg/kg), which have arid temperate continental climates (Figure 2A,B). No samples were positive for ZEA in Transylvania, Moldavia, or the Oltenia Plain regions. The highest incidence of ZEA was detected in the dry year of 2013 (6.5%; max. 3.38 µg/kg) and the rainy year of 2014 (23.5%; max. 7.05 µg/kg). No samples with ZEA incidence were found in the dry years of 2012 and 2015 (Figure 2D). The processed cereals (silo-corn germs, wheat flour type 1350) that were positive for ZEA were sampled from the Southern Plain and Dobrogea (Tandarei, Ialomita county; co-occurrence of ZEA and OTA) (Figure 4).

### 2.3. Statistical Analysis between Post-Harvest Mycotoxins and the Geographic and Agroclimatic Conditions in Romania

#### 2.3.1. Comparison of the Mycotoxin Distribution by Product Type, Agricultural Region, and Year

The distribution of AF in raw cereals and DON, OTA, and ZEA in processed cereals was greater in the Southern Plain and Dobrogea under the influence of the agroclimatic conditions in 2012–2015 (Figure 2 and Figure 3). The mycotoxin distribution by product type revealed a very significant difference in DON (significance level = 0.000, *N* = 91) (Figure 2A) and a non-significant difference in distribution for AF (significance level = 0.616, *N* = 98) in processed cereals, even though the AF contamination was very high in post-harvest raw cereal (maize) from the Southern Plain and Dobrogea in the dry years of 2012 and 2013 (Figure 3B).

#### 2.3.2. Dependence of Mycotoxin on the Agricultural Region—Correlations between Mycotoxins and Geographic Position

Because the statistical analysis of the differences between contamination levels by region revealed non-significant differences, the contamination with mycotoxins was considered to be local. Therefore, the analysis took geographic position into account, considering the locations of mills and storage units of the products (cereals, processed cereals, cereal-based foods). The graphical method was used to determine the distribution of each mycotoxin (panels on lines and columns, variables = Northern latitude, Eastern longitude, agricultural region, and year), the Baragan Plain was delimited as the area of interest between 44 and 45° N and 26 and 27° E (Figure 4). From this area, the linear correlations between mycotoxins, climatic factors, and aridity indices were determined through filtration by year as well as by the total study period in order to obtain a more substantial number of measurements.

#### 2.3.3. Correlation between Mycotoxins and Meteorological Factors

DON was inversely correlated with the cumulative precipitation in all years (rxy = −0.296 *), and it was more robustly and directly correlated with cumulative precipitation in 2012 (rxy = 0.521 *) (Table 1).

AF and ZEA were not significantly correlated with meteorological factors or aridity indices (de Martonne aridity index, Iar-dM; climatic water deficit, CWD) either over the entire study period or in filtered years (Table 2).

When the study period was taken as a whole, OTA was distinctly and significantly directly correlated with cumulative precipitation (rxy = 0.416 **) and significantly inversely correlated with the mean air temperature (rxy = −0.317 *). In 2014, OTA was directly correlated with the cumulative rainfall by year (rxy = 0.904 *, *N* = 6) in the Southern Plain and Dobrogea for the Tandarei locality (Table 2).

#### 2.3.4. Dependence of Mycotoxins on Soil Type

The dependence of mycotoxins on the soil type was analyzed by the non-parametric Jonckheere–Terpstra test for independent samples, which revealed the dependence between OTA and chernozem soils (significance level = 0.003, *N* = 59).

All of the analyzed situations showed non-significant correlations between mycotoxins and the water reserve in soil, and the aridity indices (Iar-dM, CWD) were predicted (Table 1 and Table 2).

## 3. Discussion

### 3.1. Mycotoxins in the Context of the Geographic and Agroclimatic Conditions of Romania

In Romania, post-harvest contaminations with AF, OTA, DON, and ZEA were detected in the Southern Plain and Dobrogea (Baragan Plain), under the extreme meteorological events in the period of 2012–2015. The Baragan Plain is located at latitude 44–45° N and longitude 26–27° E. It has calcic chernozem soil and a semi-arid temperate continental climate with sub-Mediterranean influences (multiannual average air temperature of 10–>11 °C, and a moderately dry and dry pluviometric regime <400 mm). This region is considered the second-most arid area after the Black Sea coast and is predicted to be the most vulnerable area to climate change by 2100 [6,25,27]. Due to its geographic position and semi-arid agroclimatic conditions, the Baragan Plain has optimal conditions for cereal contamination with *Aspergillus* sp. and AF [28,29]. However, it is also an important agricultural area of Romania, providing the highest corn production and grain storage capacity (755 authorised silos and warehouses, with a total capacity of 3,949,506 t), mainly through foreign traders.

In June–August of 2012–2015, the maximum air temperature exceeded 30 °C in the Southern Plain and Dobrogea (Baragan Plain), with the highest temperatures occurring in August (44 °C in 2012, 40 °C in 2013, 38 °C in 2014, and 41 °C in 2015). Hot summers in Southern parts place Romania in the group of Carpathian countries (Croatia, Hungary, Slovakia, Czech Republic, Poland, Ukraine, Romania, and Serbia) that are subject to heat wave events [30]. Among the cereals analyzed for the period of 2012–2015, maize was the most affected in terms of both production (−28% in the dry year of 2012, relative to the period of 1961–2016) and contamination with AF, followed by feed contamination with AF and cow’s milk contamination with AFM1. Despite the drought in 2012, Romania remained a significant cereal producer at the global level (first group for maize, second group for triticale and soybeans, third group for sorghum) [7].

In-field cereal contamination with DON is frequent in the Western and Northern regions where *F. graminearum* and *F. culmorum* are predominant, while it is sporadic and low in the Southern and Eastern areas [10,11,12,13,14,15,17,26,31]. Although there were reports of high DON contamination in grains during harvest in 2012, 2013, and 2014 [17,26], the present post-harvest study found low contamination, except in wheat flour type 450 obtained from wheat in 2014. The fact that 2014 was very rainy caused cereal control to be tightened at its reception at warehouses to prevent mycotoxin from entering the food chain. Post-harvest contamination with DON showed a very significant correlation with the processed cereals, and a significant correlation with the cumulative precipitation in all years of the study period. A very significant correlation of DON with meteorological factors (air temperature, precipitation, relative humidity) and regions was reported only for in-field contaminated cereals [32,33]. The use of modern cereal processing technologies by large companies in Romania can reduce contamination with DON up to 73–75%, even in wet sub-Carpathian regions where wheat mills are located [18]. Between 2012 and 2015, DON contamination was not detected in any cereal-based food sample.

In-field and post-harvest cereal contamination with AF is more frequent in the Southeastern part of the Southern Plain and Dobrogea (Baragan Plain), which is characterized by an arid temperate continental climate and chernozem soils that require irrigation to maintain high fertility. The agroclimatic conditions of the region vary annually and favor contamination with both *Aspergillus* sp. (*A. flavus*, *A. fumigatus*) and *Fusarium* sp. (*F. graminearum*, *F. culmorum*), with maize being the most contaminated grain [15]. The extreme weather events (high temperatures, drought) in Romania in 2012 and 2013 favored in-field maize contamination with AF and its transfer to cow’s milk as AFM1, which were noted in the Rapid Alert System for Food and Feed (RASFF) [34,35,36]. Post-harvest contamination with AF showed a non-significant correlation with meteorological factors and aridity indices, and a non-significant distribution in processed cereals, even if it was very high in post-harvest raw cereals (maize) from the Southern Plain and Dobrogea in the dry years of 2012 and 2013.

In-field and post-harvest contamination with OTA were reported only in the Southeastern part of the Southern Plain and Dobrogea (Baragan Plain) in co-existence with AF contamination in corn germs under the influence of extreme drought in 2012 and 2013. Post-harvest contamination with OTA showed an inversely significant correlation with the mean annual temperature, a significant correlation with the cumulative precipitation in all years, and, finally, a dependence on chernozem soils with high aridity indices. In Romania, cereal contamination with OTA has been reported in both the Southeastern region, which features a semi-arid temperate continental climate, as well as in the Western region, with has a semi-wet temperate continental climate [11,12,15].

Post-harvest contamination with ZEA was very low in the period of 2012–2015, but it was detected both in regions with a wet temperate continental climate (Transylvania, Southern Hilly Area) and regions with an arid temperate continental climate (Southern Plain and Dobrogea, Moldavia). Post-harvest contamination with ZEA was non-significantly correlated with agrometeorological factors (air temperature, precipitation, soil water reserve, aridity indices) when the analyzed years were taken as a whole. A significant correlation of ZEA with meteorological factors has been reported for in-field contaminated cereals [11,12,13,14,31,32].

The distribution of mycotoxins by product type was found to be influenced by the agricultural region and meteorological factors in Romania, in 2012–2015. The highest distribution of mycotoxins was recorded in the Southern Plain and Dobrogea, which both experienced drought in 2012 and 2013 and abundant rainfall in 2014. The existence of certified quality control systems in companies has limited mycotoxin contamination at the initial stages of the cereal chain (raw cereal, processed cereal), so that the cereal-based food produced is safe for consumption [18]. The non-significant correlation between post-harvest contamination with mycotoxins and agroclimatic factors can be explained by the control of cereal batches at the warehouse reception, the modernization of old or construction of modern storage spaces, and the implementation of high-performance processing technologies and quality management systems in agri-food companies. Romania has been a member of the European Union since January 2007, and must comply with the regulations for cereal storage and contamination of agri-food products with mycotoxins. We believe that these criteria have a greater contribution to the post-harvest non-significant correlation between mycotoxins and climatic factors, as compared with the ecosystem inside silos and the complex interactions between abiotic and biotic factors [37,38].

### 3.2. Mycotoxins in the Context of Current and Future Climate Conditions in Southeastern Europe

In 2012–2015, extreme weather events were recorded in Southeastern Europe (Romania, Serbia, Croatia, Macedonia, Albania) with drought in the years of 2012, 2013, and 2015, and abundant rainfall and floods in the year, 2014 [39,40]. Although the air temperature and precipitation showed significant differences between the years, this period was included in the warmest 15 years at the global level, compared to the reference period of 1961–1990 [40,41]. 

In 2012 and 2013, cereals contaminated with AF were redirected to animal feed, leading to the contamination of cow’s milk with AFM1 and RASFF notification of feed corn contaminated with AF. These incidences also led to numerous scientific publications [34,35,36,42,43,44,45,46,47]. In 2014, cereal contamination with DON was reported in the Northern part of Southeastern Europe, in Romania and Serbia [26,47]. In the period of 2012–2015, RASFF published a total of 1478 notifications on mycotoxins in food and feed, most frequently for AF (86.1%) in the dry years of 2012, 2013, and 2015, for OTA (11.2%) in the dry year of 2013 and the rainy year of 2014, and for both DON (1.4%) and fumonisin B (FB) (1.3%) in the rainy year of 2014 [34,48,49]. Cereal contamination with OTA in both dry and rainy years occurred because the fungus grows under different agroclimatic conditions; *Aspergillus* sp. is predominant in warm and temperate regions, and *Penicillium* sp. is predominant in colder areas [50,51].

In Southeastern Europe (Southern Romania, Northern Bulgaria, Northern Serbia, Eastern Croatia), the most fertile agricultural areas are plains with chernozem soils, on which wheat and maize crops are mainly grown. Because of their structure, chernozems are heavily affected by extreme weather events and climate change. In this European subregion, heat and drought events have intensified and led to a high-risk climate index (51–100), apart from Romania, which has a medium climate risk (21–50) and the highest aridization trend in Baragan Plain [6,23,24,30,52].

Knowing the current and future agroclimatic conditions is very important for the prevention and control of mycotoxins throughout the agri-food chain (in-field and post-harvest) because the climate change scenario in Southern Europe has involved an increase in the frequency, intensity, and duration of extreme weather events (drought, floods), as well as temperature increases of 1.8 °C–2.1 °C and a decrease in rainfall by 6% [24]. These climatic changes will influence fungal growth and mycotoxin production, depending on the ability of fungi to adapt [29,53,54,55,56]. Climate scenarios for the Balkans have shown that maize will be most affected by AF contamination in the +2 °C scenario; this situation is also valid for the Southern part of Romania [57,58].

In Europe, cereal contamination with mycotoxins varies depending on the geographic position and agroclimatic conditions. For example, AF, OTA, and FB prevail in the Southern region, which has a hot/wet Mediterranean climate, while DON and ZEA prevail in the Northern region, which has a colder climate [11,12,15,50,51,59,60,61,62,63,64,65].

## 4. Conclusions

This article provides important information on contamination with mycotoxins in the agri-food chain (post-harvest raw cereal, processed cereal, and cereal-based food) in the context of geographic and agroclimatic conditions of Romania in the period of 2012–2015, which involved extreme meteorological events. This included information regarding the current and future climate conditions of Southeastern Europe.

Post-harvest contamination with DON, AF, OTA, and ZEA mycotoxins was of a local character, with the Baragan Plain (southeast part of the Southern Plain and Dobrogea region) being the area of interest due to its agroclimatic conditions—a semi-arid temperate continental climate and chernozem soil with high aridity indices—which were strongly influenced by the annual meteorological events.

Among the cereals, maize was most affected by the extreme weather conditions, with the lowest productivity and high contamination with AF in the dry year of 2012, and high contamination with OTA in the dry year of 2013. DON and OTA were significantly correlated with the cumulative precipitation in all years, while AF and ZEA were not significantly correlated with climatic factors or aridity indices. The distribution of mycotoxins by product type and the non-robust correlations between post-harvest mycotoxins and agrometeorological factors could be explained by the use of quality management systems that control cereal production at warehouse receptions, performant processing technologies, and the high quality of storage spaces of the agri-food companies. Redirection of contaminated cereal batches to feed has led to RASFF notifications of contaminated feed and cow’s milk exports from Romania and other Southeastern European countries, which have caused financial and image crises in the economic and public health sectors.

Because this European subregion is subject to climate change (temperature rise, rainfall decrease), agri-food producers and competent authorities need to be aware of its impact on cereals and food quality and safety in order to develop an integrated management system to reduce the effects on human and animal health, as well as on the economy. This is the first time that post-harvest contamination (warehouse–processing–food) has been investigated in Romania or Southeastern Europe in general, and this approach can be applied in any country or even at the continental level along the agri-food chain.

## 5. Materials and Methods

### 5.1. Post-Harvest Samples and Product Types

Between January 2012 and July 2015, samples were randomly collected (*N* = 152; 1 kg/sample) from private cereal traders and food manufacturers with work points in five of the six agricultural regions of Romania (the Southern Plain and Dobrogea; the Oltenia Plain; the Southern Hilly Area; Moldavia; Transylvania; none in the Western Plain), which have certified quality systems according to the European legislation in force. Cereal-based food manufacturers also own mills from which they distribute processed cereals to their work points throughout the country. Systems of quality and distribution within and between companies form rigorous control processes and reduce the risk of mycotoxin contamination along the cereal chain.

To assess contamination with mycotoxins, the samples were grouped into three product types: Raw cereals (35 samples: Wheat, maize in silos, triticale, soybean, sorghum), processed cereals (69 samples: Wheat flour and bran; maize extract and starch, corn germs), and cereal-based foods (48 samples: Bread, cakes, pasta).

### 5.2. Mycotoxin Analysis

The samples were milled with a Retsch ZM 200 ultracentrifugal mill (Retsch, Haan, Germany) and analyzed for deoxynivalenol (DON), total aflatoxins (AF), ochratoxin A (OTA), and zearalenone (ZEA) by the enzyme-linked immunosorbent assay (ELISA) coupled with a *S*unrise microplate reader (absorbance 450 nm; Tecan, Salzburg, Austria). Mycotoxin analysis was performed with Ridascreen^®^ DON, Ridascreen^®^ Aflatoxin Total, Ridascreen^®^ Ochratoxin A 30/15, and Ridascreen^®^ Zearalelon test kits (R-Biopharm, Darmstadt, Germany). The ELISA laboratory used in the current study is accredited for the analysis of mycotoxins in cereal and food samples according to ISO 17025:2005 (by the Romanian Accreditation Association (RENAR)) and participated in the FAPAS Proficiency Testing Programme (FERA, Sand Hutton, NY, UK) with satisfactory results, demonstrating laboratory performance and results comparable to laboratories worldwide that use different methods. 

The 152 post-harvest samples were analyzed for the presence of mycotoxins (total of 331 tests: DON 91; AF 97; OTA 59; ZEA 84) depending on the product type and according to the Commission Regulations (EC) no. 1881/2006 and no. 1126/2007 [66,67], which set the maximum levels of contaminants as follows:−DON: Cereal—1250 µg/kg; processed cereal—750 µg/kg; food (bread, cakes)—500 µg/kg;−AF: Cereal—4 µg/kg; processed cereal—4 µg/kg; silage maize—10 µg/kg;−OTA: Cereal—5 µg/kg; processed cereal—3 µg/kg; and−ZEA: Cereal—100 µg/kg; processed cereal—75 µg/kg; food (bread, cakes)—50 µg/kg.

### 5.3. Geographic Coordinates

The Northern latitude and Eastern longitude of each county were determined using Google Earth [68], and counties were grouped into agricultural regions, based on their agroclimatic factors.

### 5.4. Agroclimatic Data

Agrometeorological factors (air temperature, precipitation, soil moisture reserve) were recorded by the official agrometeorological network with Meteorological Automatic Weather Station (MAWS) and Ceres-Wheat and Decision Support System for Agrotechnology Transfer (DSSAT) v.3.5. software between 1 September 2011 and 31 August 2015. The dominant soil types (chernozem, phaeozem, luvisol) in each county were set at a scale of 1:1,500,000 based on the Soil Atlas of Europe [8]. The aridity indices (de Martonne aridity index, Iar-dM; climatic water deficit, CWD) of each county were set based on data published for the period of 1900–2000 [6] to determine the correlation with mycotoxins on a long-term basis.

### 5.5. Data Collection

All data were collected in an Excel file and transferred into an SPSS database with the following variables: Years (2012–2015, with extreme meteorological events), mycotoxins (DON, AF, OTA, ZEA), post-harvest product types (raw cereal, processed cereal, cereal-based food), geographic coordinates (Northern latitude, Eastern longitude, agricultural region), and agroclimatic data (agrometeorological factors—air temperature, precipitation, and soil moisture reserve; soil types—chernozem, phaeozem, and luvisol; aridity indices—Iar-dM and CWD).

### 5.6. Statistical Analysis

The influences of the geographic position and agroclimatic conditions on post-harvest contamination with mycotoxins in Romania were determined by statistical analysis using SPSS v.23 software (IBM, Armonk, NY, USA) (Statistical Package for the Social Sciences software with ANOVA, general linear model, Pearson correlation, statistical tests for means comparison, non-parametric Kruskal–Wallis, and Jonckheere–Terpstra tests, graphical methods). The probability was considered to be statistically significant at *p* ≤ 0.05. The comparison of agrometeorological factors by region and year was made for the March–August period because there were differences in this period that influenced the growth of cereal crops; for example, the spring season began very early in the south, starting in February–March. The statistical analysis between agrometeorological factors and mycotoxins was performed for the critical periods specific to wheat and maize (May–June for DON; July–August for AF, OTA, and ZEA).

### 5.7. Replicability and Results of this Approach

The replicability and results regarding post-harvest contamination with mycotoxins are supported by data on the in-field contamination with mycotoxins (DON, AF), as well as the physico-chemical quality and productivity of annual crops (wheat, rye, triticale) under the influence of geographic position and agroclimatic conditions in Romania, between 2012 and 2016. All data for the in-field cereals (≈3600 samples) were statistically analyzed by SPSS v.23 software and evaluated in the context of the climate conditions for Romania and climate change in Southeastern Europe by the year, 2100. It is anticipated that these data will contribute to the understanding of mycotoxin occurrence in the geo-agroclimatic context of this European subregion (manuscripts are being drafted).

## Figures and Tables

**Figure 1 toxins-10-00533-f001:**
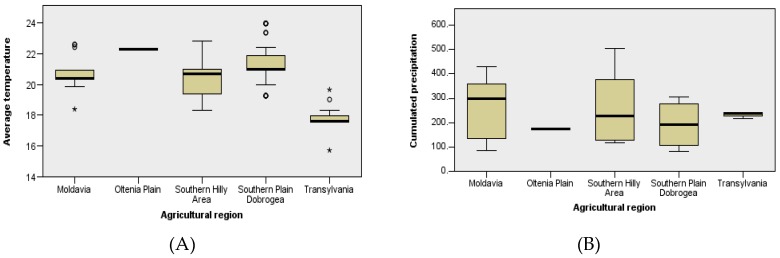
Comparisons of the distribution of agrometeorological factors by agricultural region and year: Air temperature (**A**), cumulative precipitation (**B**), soil moisture reserve (**C**,**D**) (Kruskal–Wallis non-parametric test for independent samples). °, °°, *°, ****°, *°°°°°—position of the extreme individual values.

**Figure 2 toxins-10-00533-f002:**
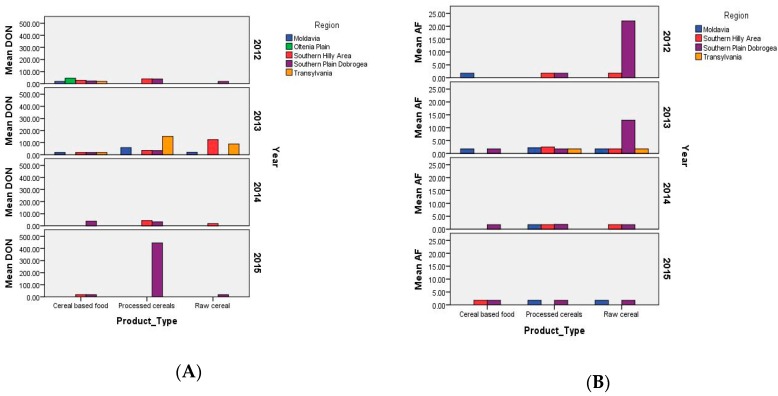
Distribution of deoxynivalenol (**A**), total aflatoxins (**B**), ochratoxin A (**C**), and zearalenone (**D**) by product type, agricultural region, and year in Romania in 2012–2015 (graphical method).

**Figure 3 toxins-10-00533-f003:**
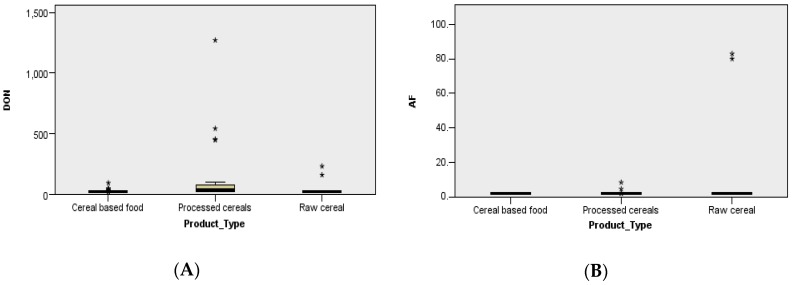
Distribution of deoxynivalenol (**A**) and total aflatoxins (**B**) by product type in Romania in 2012–2015 (Kruskal–Wallis non-parametric test for independent samples). * and **—position of the extreme individual values.

**Figure 4 toxins-10-00533-f004:**
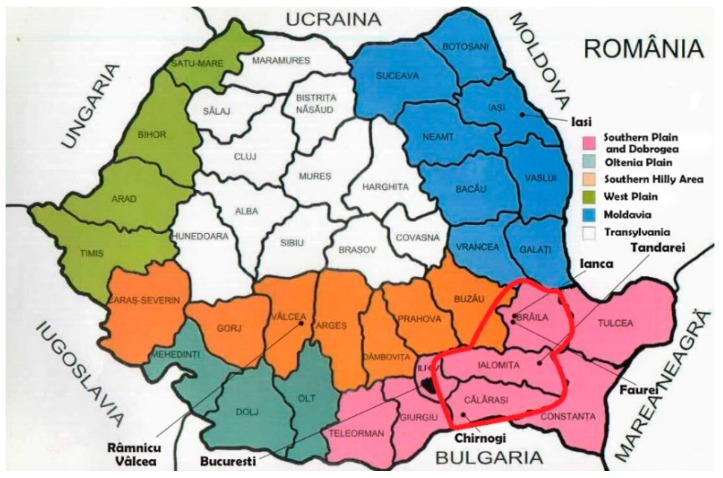
The geographic origin of post-harvest samples with mycotoxins above the maximum level in Romania, in 2012–2015. DON—Calarasi county; AF—Braila, Ilfov, Valcea, and Iasi counties; OTA—Ialomita and Braila counties. Red shape: Baragan Plain (44–45° N, 26–27° E) in the Southern Plain and Dobrogea region.

**Table 1 toxins-10-00533-t001:** Correlations between mycotoxins (deoxynivalenol (DON)) and the geographic position of the Baragan Plain, aridity indices, and agrometeorological factors by year in Romania in 2012–2015 (Pearson correlation coefficient).

Correlations for the Baragan Plain (44–45° N, 25–27° E)		Annual Agrometeorological Factors, 2012–2015			May–June Agrometeorological Factors, 2012–2015		Aridity Indices, 1900–2000	
		Cumulative Precipitation	Average air Temperature	Soil moisture Reserve	Average air Temperature	Cumulative Precipitation	De Martonne (Iar-dM)	Climatic Water Deficit (CWD)
**DON All Years**	Pearson Correlation	−0.296 *	−0.028	−0.040	0.120	−0.195	0.003	−0.029
	Significance (two-tailed)	0.035	0.844	0.779	0.404	0.169	0.983	0.837
	*N*	51	51	51	51	51	51	51
**DON 2012**	Pearson Correlation	0.521 *	0.092	−0.277	0.353	0.279	0.059	0.315
	Significance (two-tailed)	0.022	0.709	0.250	0.139	0.247	0.811	0.189
	N	19	19	19	19	19	19	19
**DON 2015**	Pearson Correlation	0.619	−0.057	−0.215	−0.435	0.801 *		−0.573
	Significance (two-tailed)	0.102	0.893	0.608	0.281	0.017		0.138
	*N*	8	8	8	8	8	8	8

*—correlation is significant at the 0.05 level (two-tailed).

**Table 2 toxins-10-00533-t002:** Correlations between mycotoxins (aflatoxins (AF), ochratoxin A (OTA), zearalenone (ZEA)) and the geographic position of the Baragan Plain, aridity indices, and agrometeorological factors by year in Romania in 2012–2015 (Pearson correlation coefficient).

Correlations for the Baragan Plain (44–45° N, 25–27° E)		Annual Agrometeorological Factors, 2012–2015			July–August Agrometeorological Factors, 2012–2015		Aridity Indices, 1900–2000	
		Cumulative Precipitation	Average air Temperature	Soil moisture Reserve	Average air Temperature	Cumulative Precipitation	De Martonne (Iar-dM)	Climatic Water Deficit (CWD)
**AF All Years**	Pearson Correlation	−0.021	0.064	−0.142	0.025	−0.125	−0.070	−0.062
	Significance (two-tailed)	0.876	0.642	0.296	0.856	0.357	0.608	0.652
	*N*	56	56	56	56	56	56	56
**OTA All Years**	Pearson Correlation	0.416 **	−0.317 *	−0.112	−0.156	0.107	−0.094	0.044
	Significance (two-tailed)	0.006	0.038	0.481	0.319	0.495	0.550	0.777
	*N*	43	43	42	43	43	43	43
**OTA 2014**	Pearson Correlation	0.904 *	0.329	0.236	0.357	0.796		0.239
	Significance (two-tailed)	0.013	0.525	0.652	0.487	0.058		0.648
	*N*	6	6	6	6	6	6	6
**ZEA All Years**	Pearson Correlation	0.217	−0.118	−0.046	−0.043	0.016	−0.136	−0.124
	Significance (two-tailed)	0.142	0.428	0.757	0.774	0.915	0.362	0.406
	*N*	47	47	47	47	47	47	47

*—correlation is significant at the 0.05 level (two-tailed); **—correlation is significant at the 0.01 level (two-tailed).

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
