# Peer review of "Post-Harvest Contamination with Mycotoxins in the Context of the Geographic and Agroclimatic Conditions in Romania"

_toxins, 2018, doi:10.3390/toxins10120533_

Reviewer 1 Report

The authors apply an ELISA based method to evaluate the presence of multiple mycotoxins on three post-harvest products, such as raw cereal, processed cereal and cereal-based food and intend to correlate data of mycotoxins contamination with Romanian climatic conditions in the 2012-2015.

-          Table 1 to 3, are only about physical ambient conditions and treated statistically, please explain the relevance of this point. The same for comparisons between the averages of soil moisture reserve by region and year discussed in line 119.

-          Please explain, how the overall incidence of  DON is 42.9% and the incidence in 2014 was 5.6%

-          Line 147. This conclusion is not supported by data.

-          Please, correct the range concentrations recorded for all the mycotoxins analyzed.

-          Please explain the dependence of product type and mycotoxins detected.

-          Line 341, this data do not correspond to BIOMIN World Mycotoxin Survey 2017, please check at:

-          https://info.biomin.net/acton/attachment/14109/f-0751/1/-/-/l-0009/l-0009:97cb/MAG_MTXsurveyReport_2017_EN_0118_low.pdf

Author Response

Dear REVIEWER 1,

We would like to thank you for the time and effort in considering our manuscript ID: toxins-384559.

We are grateful for your careful comments and have tried to respond each, considering these revisions have resulted in a significantly improved manuscript.

In the revised manuscript, the changes made are highlighted (red), but the numbering lines were altered.

In attachment are our point-by-point responses.  

Yours faithfully, 

Authors

Reviewer 2 Report

The paper "Post-harvest contamination with mycotoxins in the context of geographic and agroclimatic conditions of Romania" shows an exhaustive and very interesting picture of the situation of the post-harvest mycotoxins contamination in Romania, over a period between 2012 and 2015, taking into account the influence of many factors such as climatic conditions, geografical localization, type of cultures and their final use.

Data are well analyzed, based on a robust statistical analysis, clearly presented and enough well discussed.

I only have a curiosity that does not affect my final decision. I just wonder why authors decided to analyse the different factors separately and not by performing a multivariate analysis in order to deeply investigate the effect of every single variables on the mycotoxin content.

However, it is my opinion the paper can give important information and can be accepted for publication in the present form.

Author Response

Dear REVIEWER 2,

We would like to thank you for the time and effort in considering our manuscript ID:toxins-384559.We are grateful for your careful comments and have tried to respond each, considering these revisions have resulted in a significantly improved manuscript.In attachment are our point-by-point responses. Yours faithfully, Authors

Round  2

Reviewer 1 Report

Changes were made following instructions. Corrections done. The manuscript is suitable for publication

Regards

Author Response

Dear Editors,

We would like to thank you for the time and effort in considering our manuscript ID: toxins-384559.

We are grateful for your comments and responded to each, considering these revisions have resulted in a significantly improved manuscript.

In the revised manuscript, the changes made are highlighted (red).

In attachment are our point-by-point changes.

Yours sincerely,

Valeria Gagiu
